# Evaluation of Short-Term Stability of Different Nitazenes Psychoactive Opioids in Dried Blood Spots by Liquid Chromatography-High-Resolution Mass Spectrometry

**DOI:** 10.3390/ijms252212332

**Published:** 2024-11-17

**Authors:** Alessandro Vitrano, Alessandro Di Giorgi, Vincenzo Abbate, Giuseppe Basile, Nunzia La Maida, Simona Pichini, Annagiulia Di Trana

**Affiliations:** 1Department of Analytical, Environmental and Forensic Sciences, King’s College London, 150 Stamford Street, London SE1 9NH, UK; alessandro.vitrano@kcl.ac.uk (A.V.); vincenzo.abbate@kcl.ac.uk (V.A.); 2Department of Biomedical Science and Public Health, University “Politecnica delle Marche” of Ancona, Via Tronto 10/a, 60124 Ancona, Italy; digiorgiale97@gmail.com (A.D.G.); g.basile@staff.univpm.it (G.B.); 3National Centre on Addiction and Doping, Istituto Superiore di Sanità, V. Le Regina Elena 299, 00161 Rome, Italy; nunzia.lamaida@iss.it (N.L.M.); annagiulia.ditrana@iss.it (A.D.T.)

**Keywords:** nitazenes, stability study, LC-HRMS/MS, dried blood spots

## Abstract

Nitazenes represent a new synthetic opioids sub-class belonging to new psychoactive substances (NPSs). Their high pharmacological potency has led to numerous intoxications and fatalities, even at minimum doses. The aim of this study was to assess the stability of four nitazenes (etazene, flunitazene, isotonitazene and protonitazene) in dried blood spot (DBS) samples at different storage temperatures (room temperature and 4 °C) and determine the optimal storage conditions. Moreover, we developed and validated a new and fast liquid chromatography–high-resolution mass spectrometry method by the optimization of chromatographic conditions with the use of a different chromatographic column and mobile phases. Two concentrations, 1 and 5 ng/mL, were chosen based on the available data on nitazenes-related intoxications and their stability was evaluated at days 0 (control), 1, 7 and 30. The results showed that all analytes at 1 ng/mL were not detectable after 30 days at room temperature; a similar pattern was observed for 1 ng/mL etazene and isotonitazene samples when stored at 4 °C, whereas flunitazene and protonitazene decreased to a mean of 66% and 69% initial concentrations, respectively, at day 30. Differently, all analytes at 5 ng/mL were quantified above 44% and 41% initial concentrations at room temperature and 4 °C, respectively, showing a higher stability. The study of nitazenes stability in DBSs represents an important tool to determine the optimal sample storage conditions, such as temperature and time between sample collection and analysis. In contrast to another study, our study showed distinct stability behaviors for every investigated analyte, which also depended on the concentration. Therefore, it is difficult to define an optimal storage condition acceptable for all nitazenes. Room temperature proved to be the best medium- and long-term storage conditions for the highest concentrations, but the stability of low levels of flunitazene and protonitazene improved at 4 °C.

## 1. Introduction

New psychoactive substances (NPS) represent a large class of compounds designed to reproduce the pharmacological effects of classic drugs of abuse, bypassing anti-drug laws due to differences in the chemical structures with the already scheduled illicit substances [1]. In particular, the class of new synthetic opioids (NSOs) has gained the attention of scientists and law enforcement in the last decade due to their extreme toxicity and the increasing number of related fatalities. Among those, fentanyl represents a major global threat, causing more than 70,000 yearly overdose deaths in the USA (the so-called “fentanyl epidemics” in North America) [2]. In response to this health emergency, numerous strategies and political actions were taken to accelerate the risk assessment and scheduling process for this illicit drug and its analogs. However, the void left by the regulated fentanyl analogs was rapidly filled by the newly emerged NSO sub-class of benzimidazole opioids, also called nitazenes, which emerged on the illicit market in 2019 [3]. Nitazenes’ general structure is characterized by a benzimidazole ring with an ethylamine in position 1 and a substituted benzyl group in position 2 of the ring (Figure 1).

These substances exert their pharmacological activity through binding with the µ-opioid receptors, producing analgesia, sedation, euphoria, respiratory depression and strong addiction as the main effects. A pharmaceutical company first synthesized nitazenes to investigate their analgesic effects for medicinal purposes; however, they were never approved due to the narrow therapeutic window, the significant toxic effects and the risk of abuse or addiction [4]. The first illicitly marketed nitazene was etonitazene, found as a brownish powder in the late 1960s in Italy [1]. Lately, isotonitazene has been identified in the Belgian illicit market, and this led the European Union Drugs Agency (EUDA) to formally declare this compound as the first nitazene in circulation [3]. The first isotonitazene fatal intoxication was reported in Switzerland in 2019 [5]. Nowadays, other nitazene analogs are etazene, protonitazene, butonitazene, flunitazene, etonitazepyne and etonitazepipne, which were reported in the literature as related to fatal and non-fatal intoxications [6,7]. According to the United Nations Office on Drugs and Crime (UNODC), 83 NSOs were reported in 2024, comprising eight nitazenes [8].

Nitazenes represent a challenge for analytical toxicologists for several reasons. Due to their extreme potency, low doses are usually consumed, resulting in low concentrations in biological matrices. Consequently, analytical methods should be extremely sensitive, allowing the quantification of concentrations lower than 1 ng/mL. While isotonitazene pharmacokinetics and pharmacodynamics have been studied in rats [9], other nitazenes pharmacokinetics are poorly investigated in humans. As a consequence, their identification in real intoxication cases is difficult due to the lack of knowledge of the most appropriate biomarkers. Finally, little is known about their stability in biological matrices. The degradation assessment represents an important parameter to determine the optimal storage conditions, the time between sample collection and analysis and the best sample to conduct nitazenes identification. Walton et al. investigated the stability of nine nitazenes at 10 ng/mL in blood samples. A degradation pattern was observed at room temperature after 14 days for isotonitazene and protonitazene, while the same compounds showed no degradation when stored at 4 °C [10]. Conversely, Ververi et al. [11] observed that room temperature was the best sample storage condition for dried blood spots (DBSs). DBS sampling is a microsampling technique that is raising particular interest since it is less invasive, easy to use, is less expensive compared to classic blood sampling and requires small blood volumes (10–50 µL) [12]. In the last decade, DBS sampling has increasingly been recognized in different fields, such as clinical toxicology, doping control and forensic toxicology, especially for the enhancement of analytes stability at room temperature [13].

An interesting review available in the literature examined several analytical techniques used for NSO analysis in biological matrices, such as gas chromatography–mass spectrometry (GC-MS), gas chromatography–tandem mass spectrometry (GC-MS/MS), liquid chromatography–mass spectrometry (LC-MS), liquid chromatography–tandem mass spectrometry (LC-MS/MS) and liquid chromatography–high-resolution mass spectrometry (LC-HRMS) [14]. Dedicated quantitative methods for isotonitazene, metonitaneze, N-pyrrolidino etonitazene or N-piperidinyl etonitazene quantification in urine, whole blood and/or vitreous humor were developed and validated in LC-MS/MS [15,16,17,18], while a comprehensive method, to quantify eight nitazenes opioids and four related metabolites in whole blood simultaneously, was developed and validated in LC-MS/MS by Walton et al. [10] Furthermore, the etonitazepipne quantification in post-mortem samples was conducted in a dedicated HPLC-HRMS/MS method [7]. Other analytical techniques, such as GC-MS and LC-HRMS/MS, were mostly used for the structural characterization of laboratory-synthesized nitazenes or unknown metabolites [18,19].

The aims of this study were to develop and validate an analytical method in LC-HRMS/MS for the quantification of four nitazenes (etazene, flunitazene, isotonitazene and protonitazene, (Figure 2) in DBSs and to assess their stability at 4°C and room temperature, over a 30-day period. Moreover, we sought to investigate the optimal storage conditions for nitazene analogs when collected as DBSs. 

## 2. Results

### 2.1. Method Optimization and Validation

The liquid chromatography conditions were optimized with respect to a previous UHPLC-HRMS/MS method developed for the determination of etonitazepipne in post-mortem biological matrices [7] by selecting a different chromatographic column and testing different mobile phase compositions. Moreover, to enlarge the compounds list of nitazenes with respect to the old method, which included only etonitazepipne, the elution gradient was improved to enable baseline separation of the four nitazenes under investigation. To this concern, the Accucore™ Phenyl Hexyl (100 × 2.1 mm, 2.6 µm, ThermoFischer Scientific, Whaltm, MA, USA) and the FORCE Biphenyl columns (50 × 3.0 mm, 3 µm, Restek Corporation, Bellefonte, PA, USA) were tested. Although a good baseline separation and peak shape were observed with both columns, the FORCE Biphenyl one was chosen based on a short run time with a satisfactory obtained resolution. Moreover, considering isotonitazene and protonitazene, which present the same [M+H]^+^ exact mass and a similar fragmentation pattern, different mobile phase compositions were investigated during method development. Aqueous 0.1% formic acid (mobile phase A, MPA) and 0.1% formic acid in acetonitrile (mobile phase B, MPB) proved to be the best mobile phase composition for the baseline resolution of isotonitazene and protonitazene with respect to ammonium formate 2 mM, 0.1% formic acid (MPA) and ammonium formate 2 mM in methanol/acetonitrile 50:50, 0.1% formic acid (MPB). Finally, the elution gradient was optimized to allow the best chromatographic separation and acceptable reproducibility in a short time. Figure 3 shows representative chromatograms of the four nitazenes under investigation.

Mass spectrometric conditions were optimized by direct infusion of working standards of each analyte in the heated electrospray ionization (HESI) source. Once the exact mass at [M+H]^+^ was obtained, increasing collision energies were applied to select the best fragmentation pattern for each compound. The best fragmentation pattern was obtained at 70 normalized collision energy (NCE) for all the compounds. Finally, the HESI source parameters, such as the capillary voltage and temperature, were optimized by injecting a pure standards mixture in MPA/MPB 80:20 *v*/*v* into the chromatographic system to gain the optimal ionization of the analytes and increase the sensitivity of the method. Mass spectra of all the analytes under investigation are reported in Figure 4. Different extraction solvents were investigated for sample preparation, such as acetonitrile, acetonitrile/methanol 1:1 (*v*/*v*), isopropanol, chloroform/isopropanol 9:1 (*v*/*v*) and methanol; this latter provided the highest recovery percentages.

The newly developed analytical method was validated according to the Organization of Scientific Committee (OSAC) for Forensic Sciences guidelines [20]. Specifically, the method proved to have good sensitivity, with a limit of detection (LOD) of 0.25 ng/mL for etazene and flunitazene, whereas for isotonitazene and protonitazene, it was 0.5 ng/mL; the limit of quantification (LOQ) was 0.5 ng/mL for all the analytes. Linearity was assessed in the range of 1–20 ng/mL, also confirmed by the Mandel test [21]; bias and precision were within the acceptable criteria. No carryover or interferences were observed during the validation experiments. The recovery and matrix effects were within the range of 84–117% and 80–112%, respectively. All the validation parameters are reported in Table 1.

### 2.2. Nitazenes Stability

The four nitazenes under investigation showed different degradation patterns depending on the temperature storage and concentration. Specifically, all 1 ng/mL analytes were below the LOQ at 30 days when stored at room temperature. Etazene and isotonitazene also presented the same degradation pattern at 4 °C, whereas flunitazene and protonitazene proved to be more stable. Indeed, these latter decreased to a mean of 66% and 69% initial concentrations, respectively, after 30 days. A different pattern was observed for the 5 ng/mL analytes. In this case, etazene, flunitazene, isotonitazene and protonitazene were quantified at means of 44%, 99%, 95% and 90% initial concentrations, respectively, after 30 days at room temperature; when stored at 4 °C, etazene, flunitazene, isotonitazene and protonitazene were assessed at means of 55%, 93%, 41% and 45%, respectively. A high variability was observed within each group of samples, as shown in Figure 5 and Figure 6 through standard deviation bars.

## 3. Discussion

The developed and validated analytical method proved to be suitable for the analysis of the four nitazenes under investigation. It also allowed the separation of isomers isotonitazene and protonitazene, which only differs for the -isopropyl (isotonitazene) and -propyl (protonitazene) bound to the oxygen in the benzene ring. 

In this study, DBSs were investigated as a promising microsampling technique in the pharmacology and toxicology fields due to its numerous advantages; indeed, this microsampling technique is easy to use, cheap, non-invasive and requires a low volume of matrix. Furthermore, DBS sampling does not necessitate special equipment or trained healthcare professionals for sampling and also allows the collection of samples on the roadside, at crime scenes and during workplace drug testing. On the other hand, DBSs also present some disadvantages. Among these, the low collected blood volume requires sensitive analytical methods for the detection of substances. Moreover, the hematocrit effect may affect the spreading of capillary blood in the DBS card [22]. Sample collection can also be challenging due to the time for drying blood before storage (about 3 h). 

During the method’s development, particular attention was given to the internal standard (ISTD) incorporation into DBSs. Indeed, this step is crucial for compensating for any variability during the analytical process and for the quantification of the analytes under investigation. Due to the unavailability of deuterated nitazenes standards, different stable isotope standards were tested to achieve the best validation parameters. In this context, fentanyl-d_5_ was chosen as the ISTD due to its similar physicochemical properties with nitazenes and the intermediate retention time between the analytes under investigation. Furthermore, several alternatives are available for the ISTD incorporation into DBSs, such as spraying before extraction or precoating the DBS with the ISTD [23,24]. However, these techniques are not available in most laboratories and require high costs. For this reason, we decided to add the ISTD to the methanolic solution before extraction, as reported in many DBS-based analytical methods [25,26]. 

The proposed investigation represents one of the first studies on nitazenes stability in DBS samples; indeed, few studies are available in the literature. Ververi et al. [11] investigated the stability of nine nitazenes in DBSs at −20 °C, 4 °C and room temperature over 28 days. Three concentrations were examined (1, 10 and 50 ng/mL); the results showed that room temperature was the best storage condition over 28 days, with the lowest variability. Different concentrations and different time points were studied in the present research, which was chosen considering the high nitazenes pharmacological potency and the low concentrations usually detected in biological samples of consumers [7]. For this reason, higher concentrations were not evaluated. In this study, the results highlighted that 1 ng/mL of nitazenes was not detectable after 30 days when stored at room temperature; this suggests performing analyses of DBS samples within 7 days from the samples’ arrival. Differently, an increase in stability was observed depending on the concentration; indeed, 5 ng/mL samples proved to be less unstable, even at room temperature. However, considering the unknown content of the samples, it is suggested to perform an analysis within the first week if stored at this condition. Etazene and isotonitazene at 1 ng/mL were not detected at day 30, even when stored at 4 °C, whereas flunitazene and protonitazene presented a concentration less than half of the initial concentration. As expected, 5 ng/mL showed higher stability; however, an analysis should be performed within 1 week of the sample’s arrival and also when stored at a refrigerated temperature.

The stability of nitazenes in blood samples was assessed by Walton et al., investigating three storage conditions (−20 °C, 4 °C and room temperature) at 10 ng/mL. Isotonitazene, protonitazene and flunitazene proved to be stable at 4 °C over 60 days, whereas isotonitazene and protonitazene were observed to be unstable at room temperature after the fourteenth day of storage [10]. This higher stability may be a consequence of several factors. First, we observed that the degradation pattern of nitazenes was affected by the concentration; thus, the higher stability may be a consequence of the higher tested concentrations. Then, Walton et al. performed the study in 500 µL blood samples; besides the differences between the blood samples and DBSs, the higher volume may contribute to the stability of these analytes. 

The obtained results showed a high variability of ratios in some sample groups. Since the method showed satisfactory validation parameters according to the OSAC guidelines criteria, interference from the ISTD or method imprecision could be excluded. A possible reason for the variability may be the hematocrit, as reported by Deprez et al. [27], who observed a bias in the results during the determination of four immunosuppressant drugs in DBSs due to the hematocrit. Unfortunately, a comparison with similar studies is difficult since data on the intra-group variability were not reported [11], suggesting that further studies on the hematocrit effect on nitazenes’ stability in DBSs need to be performed. However, the use of suitable sample collection devices or kits may overcome this issue.

Stability studies play a crucial role in pharmacotoxicological laboratories; indeed, comprehensive knowledge of the degradation pattern of analytes in specific matrices is required for adopting the best practices. Notably, we demonstrated that etazene, flunitazene, isotonitazene and protonitazene were not detectable after 30 days when stored at room temperature. This can help to interpret the results in cases involving the analysis of nitazenes in DBS samples and to avoid false-negative results.

## 4. Materials and Methods

### 4.1. Chemicals and Reagents

Flunitazene, etazene, protonitazene, isotonitazene and fentanyl-d_5_ analytical standards were obtained from Cayman Chemicals (Ann Arbour, MI, USA). LC-MS grade acetonitrile, methanol, water and analytical grade formic acid were supplied from Carlo Erba (Cornaredo, Italy). QIAGEN QIAcard FTA DMPK DBS cards were purchased from Fisher Scientific (Hampton, NH, USA). 

### 4.2. Calibrators and Quality Control (QC) Samples

The working standard solutions for the calibrators containing the nitazenes under investigation at 30, 120, 180, 360 and 600 ng/mL were prepared by the appropriate methanolic dilution of stock solution. Similarly, the working standards for the quality control (QC) samples were prepared at 90, 240 and 480 ng/mL for the low-, medium- and high-QCs, respectively.

Fentanyl-d_5_ solution at 180 ng/mL was prepared by the appropriate methanolic dilution of stock solutions. 

Pooled drug-free human blood was obtained by 7 different volunteers and used for the preparation of calibrators and QC samples. Considering the capacity of the DBS cards, 30 µL of blank blood was deposited on each DBS and fortified with 1 µL of the corresponding working standard solution to obtain the 5 calibrators (1, 4, 6, 12 and 20 ng/mL). 

The low-, medium- and high-QC samples were set at 3, 8 and 16 ng/mL and were prepared by adding 1 µL of the corresponding working standard solution.

### 4.3. Stability Study Design

The blank, 1 and 5 ng/mL DBS samples were used for the stability study. Specifically, the blank samples were prepared by depositing 30 µL of drug-free human blood on the DBS card without the addition of working standards. Then, 1 µL of 30 ng/mL working standard solution was added to a blank DBS for the preparation of 1 ng/mL sample; similarly, 1 µL of 150 ng/mL working standard solution was added to a blank DBS to obtain the 5 ng/mL sample. Final concentrations were chosen to fit the usual nitazenes concentrations in this matrix. 

Analyte stability was assessed at different storage temperatures (4 °C and room temperature) over time (1, 7 and 30 days). Short-term stability (1 day at 4 °C and room temperature) was determined to simulate different conditions of sample transportation. The medium- and long-term stability were evaluated at 14 and 30 days, respectively, at 4 °C and room temperature. DBSs prepared on day 0 were used as control samples to normalize the instability.

### 4.4. Sample Preparation

Each circle on the DBS card was hand-cut with scissors and transferred to a clean glass tube. Then, the DBS was added with 500 µL of methanol and 2 µL of the ISTD. The analytes extraction was conducted by sonicating for 30 min and centrifuging (4000 rpm × 5 min). The methanolic solution was collected, transferred to a clean glass tube and allowed to evaporate under a gentle nitrogen stream. The dry residue was reconstituted in 30 µL of MPA/MPB 80:20 (*v*/*v*), vortexed and centrifuged (4000 rpm × 5 min). Finally, the solution was transferred to an autosampler vial before the injection of 10 µL into the LC-HRMS/MS.

### 4.5. Instrumental Analysis

Instrumental analyses were performed on a DIONEX UltiMate 3000 liquid chromatographer (Thermo Scientific, Waltham, MA, USA) coupled to a Q Exactive Focus quadrupole-Orbitrap high-resolution mass spectrometer equipped with a heated electrospray ionization (HESI) source (Thermo Scientific, Waltham, MA, USA). The chromatographic separation was carried out through a FORCE Biphenyl column (50 × 3.0 mm, 3 µm) by Restek Corporation (Bellefonte, PA, USA). Aqueous 0.1% formic acid and 0.1% formic acid in acetonitrile were MPA and MPB, respectively. The gradient elution was set as follows: 2% MPB was held for 1 min and increased to 20% until 2.5 min; then, MPB reached 95% until 8 min, prior to restoring the initial conditions (2% MPB) in 2 min, until the end of the chromatographic run. The total run time was 10 min.

The MS source parameters were set as follows: aux gas flow rate at 3, heated at 50 °C; capillary temperature, 250 °C; spray voltage, 3.50 kV. The MS acquisition mode was a FullMS/data-dependent MS/MS scan (FullMS/ddMS^2^). The full scan resolution was 70,000 *m*/*z*, the scan range was set at 120–1000 *m*/*z*, the automatic gain control (AGC) target was at 1 × 10^5^, and the maximum injection time was at 50 ms. A direct infusion of each nitazene and fentanyl-d_5_ diluted standard was performed to create an inclusion list. Specifically, the protonated adducts [M+H]^+^ were *m*/*z* 352.2383 for etazene, *m*/*z* 371.1877 for flunitazene, *m*/*z* 411.2390 for both isotonitazene and protonitazene, and 342.2588 for fentanyl-d_5_ (Table 2). The ddMS^2^ acquisition was set with an isolation window of *m*/*z* 1.5, 3 microscans and a normalized collision energy (NCE) of 70. Data were processed using Xcalibur™ (Thermo Scientific, Waltham, MA, USA).

### 4.6. Method Validation

The method’s linearity, bias, precision, sensitivity and carryover were assessed following a five-day protocol proposed by the OSAC guidelines [20]. Furthermore, the recovery and matrix effects were evaluated according to the protocol proposed by Matuszewski et al. [28].

#### 4.6.1. Linearity

The method’s linearity was evaluated in the range of 1–20 ng/mL on the first day of the five-day validation protocol; at least five calibrators were injected in triplicate. Calibrators were required to be quantified within a 15% nominal concentration, and the correlation coefficient (r^2^) was required to be ≥0.99. In addition, the Mandel test was performed to check whether the straight-line function could be used for calibration [21].

#### 4.6.2. Bias 

Bias was assessed by fortifying 3 separate blanks at each QC concentration (low-, medium- and high-QCs) over 5 different runs. The samples were required not to exceed a ±20% nominal concentration.

#### 4.6.3. Precision

Each QC sample (low-, medium- and high-QCs) was analyzed in triplicate over five different runs performed on the same day and on 5 different days. Precision was evaluated in terms of the percent coefficient of variation (%CV), and ±20% was the acceptable criterion.

#### 4.6.4. Sensitivity

The LOD was assessed by spiking a blank matrix at the LOQ concentration and diluting 5-, 10- and 20-fold. The lowest concentration at which a chromatographic peak eluted within ±0.1 min of the average calibrator retention time with a signal-to-noise ratio ≥ 3 was defined as the LOD. 

The LOQ was assessed by spiking a blank matrix at the lowest non-zero calibrator. The retention time was required to be within ± 0.1 min, and the quantification was required to be a ±20% nominal concentration.

#### 4.6.5. Carryover

Blank samples were analyzed in triplicate after the highest calibrator to evaluate the carryover. Peaks within ±0.1 min of the average calibrator retention time were evaluated; if no peaks with a signal-to-noise ratio ≥ 3 eluted in this time window, the carryover was negligible.

#### 4.6.6. Recovery and Matrix Effect

Blank matrices were spiked at low-, medium- and high-QC concentrations. Three different sets of samples were prepared. Specifically, in set A, the ISTD was added before the extraction step; in set B, the ISTD was added after the extraction and before the evaporation; set C consists of neat standards reconstituted in MPA/MPB 80:20. The mean chromatographic peak area of each analyte was used for the following calculations. Recovery was assessed by dividing set B by set A, whereas the matrix effect was determined by dividing set B by set C. Both parameters were required to be within ±30%.

### 4.7. Statistical Analysis

The Dixon test [29] (*p* < 0.05) was performed to detect and exclude outliers in the replicates of each group of samples. Stability testing was assessed by performing a one-way analysis of variance (ANOVA) test. 

## 5. Conclusions

The developed and validated analytical method allowed the determination of etazene, flunitazene, isotonitazene and protonitazene in DBS samples. Furthermore, the method proved to be suitable for the short-term stability assessment of the analytes at different storage conditions (4 °C and room temperature), considering two different concentrations in the possible range usually detected in real intoxication cases (1–5 ng/mL). The stability study highlighted the influence of storage temperature, time and concentration on nitazenes’ degradation patterns. Indeed, 1 ng/mL of nitazenes showed a complete degradation after 30 days when stored at room temperature; the same degradation pattern was observed for etazene and isotonitazene at day 30 when stored at 4 °C. Differently, an increased stability was generally observed with the highest tested concentration. Our study showed distinct stability behaviors for every investigated analyte. Therefore, it is difficult to define an optimal storage condition acceptable for all nitazenes. Room temperature proved to be the best medium and long-term storage condition for the highest concentrations, but 4 °C improved the stability of low levels of flunitazene and protonitazene. Further stability studies may be carried out to investigate other parameters as the influence of hematocrit on the concentration variability. Different DBS devices could be tested for more hematocrit control. Additionally, it is necessary to assess nitazenes’ stability over extended periods.

## Figures and Tables

**Figure 1 ijms-25-12332-f001:**
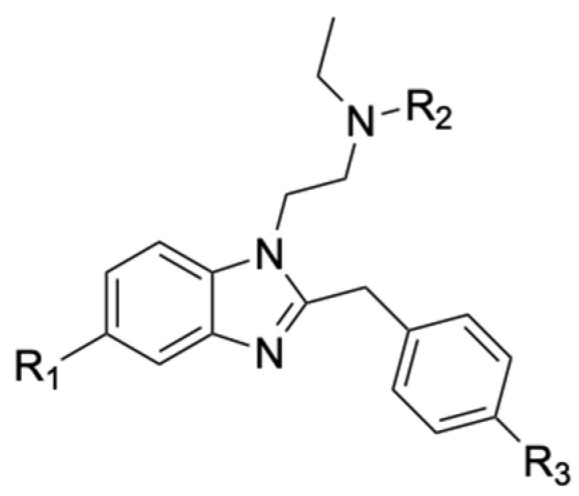
General structure of nitazenes.

**Figure 2 ijms-25-12332-f002:**
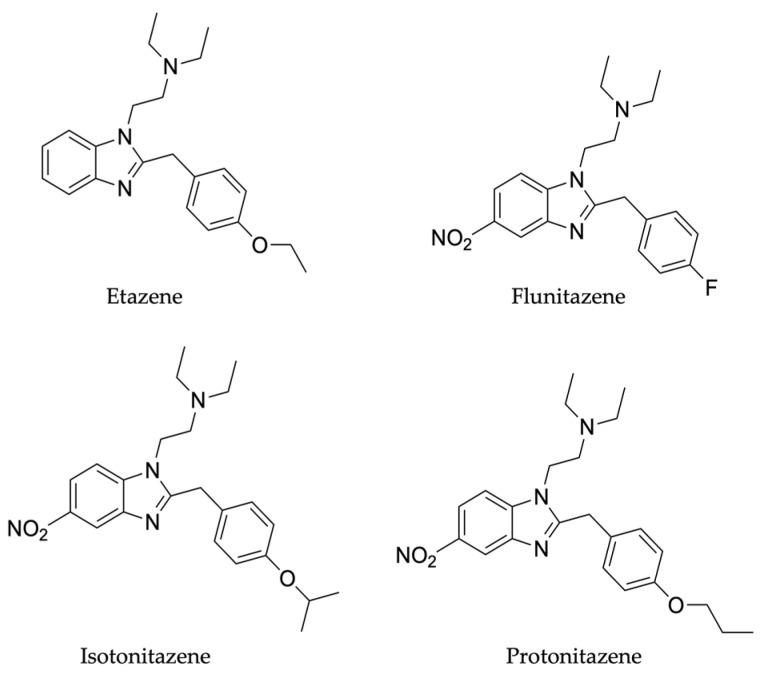
Chemical structure of nitazenes under investigation.

**Figure 3 ijms-25-12332-f003:**
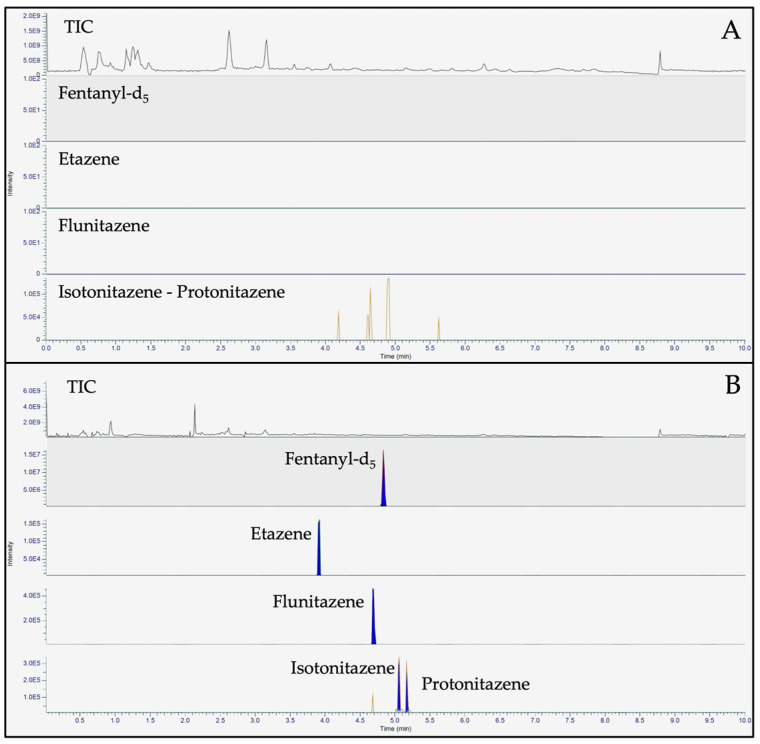
Representative TIC and EIC chromatograms of a blank sample (**A**) and a spiked sample at LOQ (**B**) of all analytes in dried blood spots.

**Figure 4 ijms-25-12332-f004:**
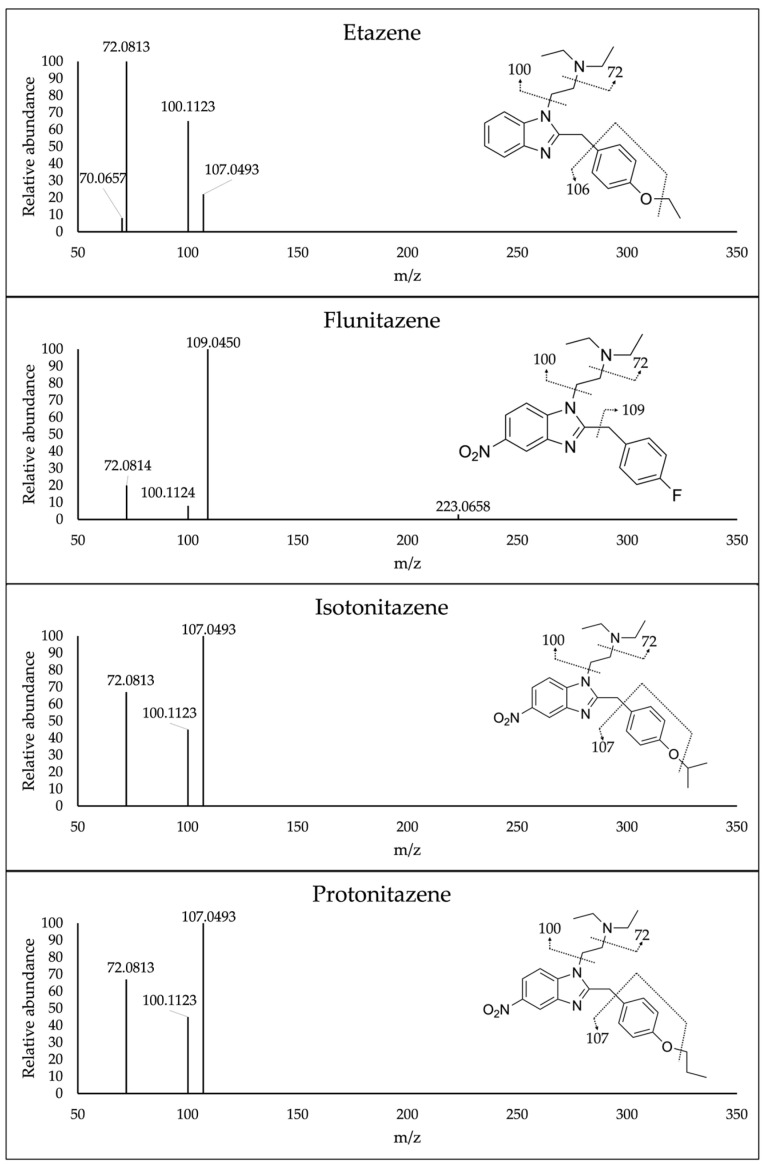
Fragmentation mass spectra of all analytes under investigation obtained at 70.00 normalized collision energy.

**Figure 5 ijms-25-12332-f005:**
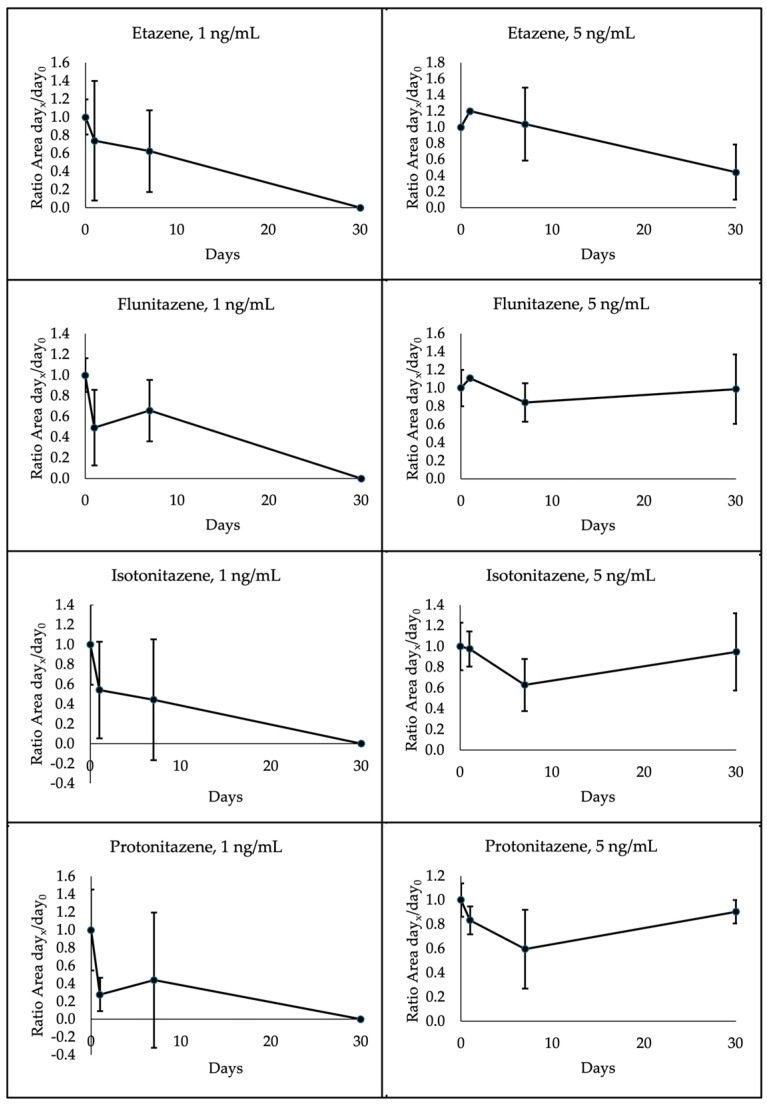
Degradation pattern of each nitazene in dried blood spots at 1 ng/mL (n = 6) and 5 ng/mL (n = 6) at room temperature over 30 days. Analyses were performed on days 0 (control), 1, 7 and 30.

**Figure 6 ijms-25-12332-f006:**
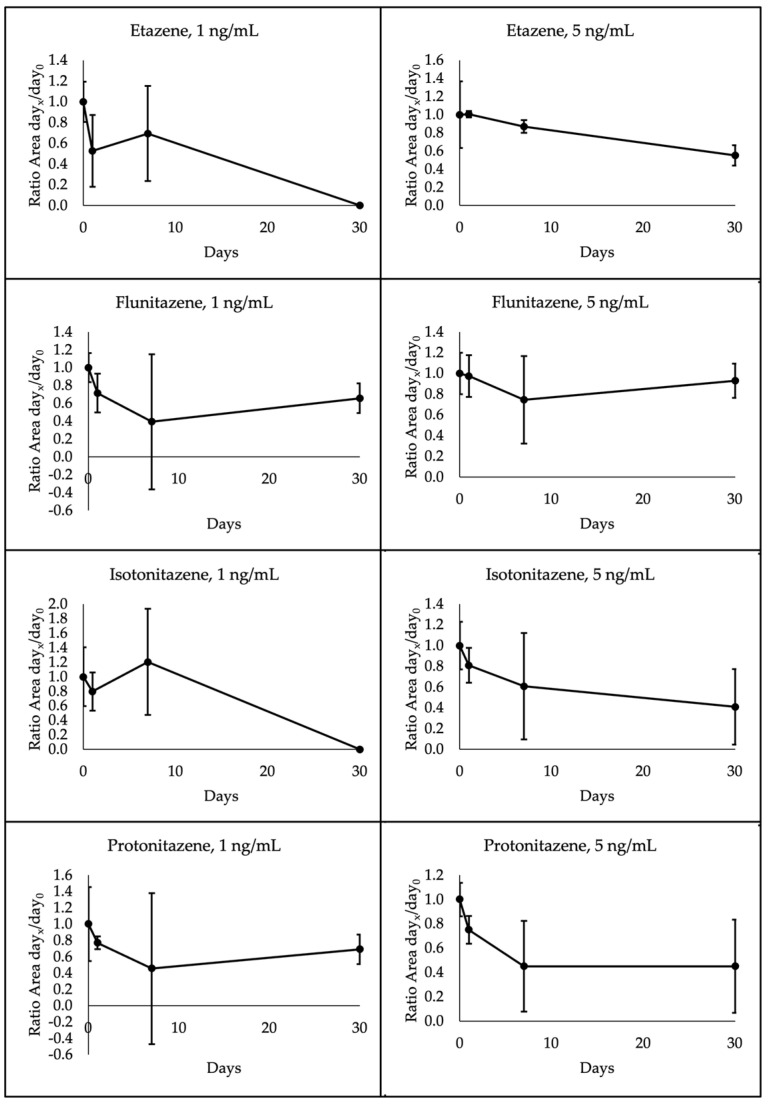
Degradation pattern of each nitazene in dried blood spots at 1 ng/mL (n = 6) and 5 ng/mL (n = 6) at 4 °C over 30 days. Analyses were performed on days 0 (control), 1, 7 and 30.

**Table 1 ijms-25-12332-t001:** Validation parameters. Low-, medium- and high-QCs were 3, 8 and 16 ng/mL, respectively.

Analyte	Linear Range (ng/mL)	r^2^	LOD (ng/mL)	LOQ (ng/mL)	Accuracy (%CV)			Inter-Day Precision (%CV)			Intra-Day Precision (%CV)			Recovery (%)			Matrix Effect (%)		
					**lQC**	**mQC**	**hQC**	**lQC**	**mQC**	**hQC**	**lQC**	**mQC**	**hQC**	**lQC**	**mQC**	**hQC**	**lQC**	**mQC**	**hQC**
Etazene	1–20	0.995	0.25	0.5	8	21	11	12	18	4	15	7	13	99	89	84	88	108	97
Flunitazene	1–20	0.997	0.25	0.5	10	9	8	15	12	9	12	9	18	95	110	105	80	110	95
Isotonitazene	1–20	0.993	0.35	0.5	17	20	12	8	16	11	18	11	12	91	117	92	108	93	96
Protonitazene	1–20	0.992	0.35	0.5	18	15	16	3	5	2	11	12	16	93	92	99	112	108	101

Abbreviations: %CV, percent coefficient of variation; hQC, high-QC; LOD, limit of detection; LOQ, limit of quantification; lQC, low-QC; mQC, medium-QC; r^2^, coefficient of determination.

**Table 2 ijms-25-12332-t002:** Retention times and [M+H]^+^
*m*/*z* (Δmass =5 ppm)of the analytes under investigation.

Analyte	[M+H]^+^ *m*/*z*	Acquisition Window (min)	Retention Time (min)
Etazene	352.2385	2.5–4.5	3.91
Flunitazene	371.1871	3.1–5.1	4.69
Isotonitazene	411.2386	4.1–6.1	5.06
Protonitazene			5.18
Fentanyl-d_5_	342.2588		4.83

## Data Availability

The data are not publicly available.

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
