# Peer review of "Evaluation of Short-Term Stability of Different Nitazenes Psychoactive Opioids in Dried Blood Spots by Liquid Chromatography-High-Resolution Mass Spectrometry"

_ijms, 2024, doi:10.3390/ijms252212332_

Round 1
Reviewer 1 Report
Comments and Suggestions for Authors
In this work, the authors developed LC-HRMS/MS method to evaluate the short-term stability of four different nitazenes in dried blood spots (DBS). Systematic investigations along with suitable data and discussion were carried out. The study of nitazenes stability in DBS represents an important tool to determine the optimal samples storage conditions, and thereby provides reasonable drug use. I would like to recommend its acceptance in International Journal of Molecular Sciences, just minor revisions.
1. Introduction, Lines 82-87, should be reorganized. The reference can be cited, but related related work corresponding to the specific method should be added, respectively. And more detailed descriptions can be added. LC-HRMS/MS hasn’t been used? Please mention clearly. If reported, also OK, you can add more detailed descriptions.
2. Conclusions. Can you outlook the presented method’s suitable for the stability evaluation? Other temperature or other amounts of samples?
Author Response
comment 1: In this work, the authors developed LC-HRMS/MS method to evaluate the short-term stability of four different nitazenes in dried blood spots (DBS). Systematic investigations along with suitable data and discussion were carried out. The study of nitazenes stability in DBS represents an important tool to determine the optimal samples storage conditions, and thereby provides reasonable drug use. I would like to recommend its acceptance in International Journal of Molecular Sciences, just minor revisions.
Response1: We would like to thank the reviewer for appreciating out study and for the suggestions
comment 2: Introduction, Lines 82-87, should be reorganized. The reference can be cited, but related related work corresponding to the specific method should be added, respectively. And more detailed descriptions can be added. LC-HRMS/MS hasn’t been used? Please mention clearly. If reported, also OK, you can add more detailed descriptions.
Response 2: In agreement with the reviewer comment, the above-reported paragraph was improved with extended information on the methods developed in each analytical techniques, clarifying all the references related to each one. (lines 92-106)
Comment 3: Can you outlook the presented method’s suitable for the stability evaluation? Other temperature or other amounts of samples?
Response 3: In agreement with the reviewer comment, the conclusion section was improved with an additional statement on the method suitability for the stability assessment of the four investigated nitazenes at different conditions (lines: 374-378).
Reviewer 2 Report
Comments and Suggestions for Authors
Although it is mentioned in the title, it should be repeated in the figures on stability that the experiments refer to dried blood samples. Furthermore, there seems to be no information about humidity and oxygen levels. I am no expert on DBS samples, but I would assume that these samples would be more stable in an extremely dry and oxygen-free atmosphere.
Furthermore, the high variability of the measurements raises some questions. The error bars are so large that no significant result could be obtained. Did the authors try to normalize the measurements for different amounts of blood? How were the DBS samples treated? Did the authors extract the entire spot or only a specific area? The use of fentanyl as an internal standard does not seem appropriate or should at least be explained in more detail, since the degradation should not be normalized in this context?
I am not convinced by the scientific content of this manuscript. It shows that most conditions are not suitable for obtaining reproducible levels of the analytes and that even at 4°C the variation is enormous. I don't understand what the reader should learn from this work, except that measuring DBS samples is difficult for these analytes. Perhaps I have not understood the message of this manuscript. At the moment, I would not recommend publishing it.
Author Response
comment 1: Although it is mentioned in the title, it should be repeated in the figures on stability that the experiments refer to dried blood samples.
Response1: Amended
comment 2: Furthermore, there seems to be no information about humidity and oxygen levels. I am no expert on DBS samples, but I would assume that these samples would be more stable in an extremely dry and oxygen-free atmosphere.
Response 2 : Dried Blood Spots is a microsampling technique for capillary blood collection that is increasingly applied in different fields of toxicology. Depending on the specific DBS device, a small volume (30-100 ml) of capillary blood is directly collected on a particular filter paper disc and dried at normal room conditions. These devices are specifically developed to enhance the stability of analytes in blood at normal condition (25°C, average humidity) and bypass the issue of sample shipping and storage. While DBS are widely used for clinical and doping purposes, information on some other analytes of interest in toxicology, especially concerning the newest substances such as nitazenes, is still to be evaluated. To better explain the importance of DBS stability assessment, we improved the introduction section (lines 86-91) with a concise and explicative paragraph on DBS utility.
comment 3: Furthermore, the high variability of the measurements raises some questions. The error bars are so large that no significant result could be obtained. Did the authors try to normalize the measurements for different amounts of blood? How were the DBS samples treated? Did the authors extract the entire spot or only a specific area?
Response 3: The DBS were prepared according to the manufacturer's instructions indicating that each circle capacity was 30ml of blood. The analyses were conducted by cutting each circle and separately preparing it according to the procedure reported in subsection 4.4 “sample preparation”. The abovementioned subsection was further improved with more detailed information for the best reader understanding. (lines 295-303)
comment 4: The use of fentanyl as an internal standard does not seem appropriate or should at least be explained in more detail, since the degradation should not be normalized in this context?
Response 4: Due to the unavailability of deuterated analogues of the investigated analytes, fentanyl-d5 was selected as the best analytical internal standard for quantification of the four nitazenes in the DBS samples. Considering the successful validation of the analytical method, we could exclude that the internal standard affected the stability assessment. Furthermore, the same internal standard was successfully used by Ververi et al. (JPBA 2024 15; 241:115975) who conducted a similar study using a different DBS card device. Conversely, the reported bias for some points was hypothesized as the effect of the different hematocrit than the control sample which was used for the normalization. To this concern, further explanation on the hematocrit effect on DBS was provided in the discussion section, highlighting the importance of hematocrit control in DBS sampling and the preference for specific devices for hematocrit control. (lines 243- 252)
comment 5: I am not convinced by the scientific content of this manuscript. It shows that most conditions are not suitable for obtaining reproducible levels of the analytes and that even at 4°C the variation is enormous. I don't understand what the reader should learn from this work, except that measuring DBS samples is difficult for these analytes. Perhaps I have not understood the message of this manuscript. At the moment, I would not recommend publishing it.
Response 5: Recently, DBS has gained particular interest from the scientific community in different fields of toxicology, due to the promising results obtained for pharmaceutical drugs and doping agents. However, these new devices suitability should be still assessed, especially for the newest analytes of interest for clinicians and laboratory staff. In this context, our results could be of particular interest. Indeed, we developed and validated an innovative analytical method in one of the most advanced hyphenated analytical techniques, such as LC-ESI-HRMS/MS which could be implemented in toxicological laboratories for the quantification of nitazenes. Furthermore, the application of the method to stability assessment of nitazenes in DBS showed that nitazenes may have different behaviour according to their structures, partially contradicting what already observed in other studies. Finally, it is important to raise the attention on the specific preventive measures such as the hematocrit homogeneity of samples, to be considered in DBS analyses, especially in forensic fields. To this concern, an extended explanation on the innovative aspects and the relevance of our results was added in the abstract, in the introduction section, discussion section and conclusion section.
Reviewer 3 Report
Comments and Suggestions for Authors
The authors assessed the stability of four opioids nitazenes (etazene, flunitazene, isotonitazene and protonitazene) in dried blood spot samples at different storage temperatures (room temperature and 4°C), using a novel well-validated liquid chromatography-high resolution mass spectrometry method. The work is well appreciated. The article is well written in a clear language. However, many revisions are required. Particularly, the authors should highlights novelty in LC-HR-MS/MS in the abstract. Besides, different conclusions should be heighted more if compared to outcomes of ref [10].
1- Title should contain the word psychoactive opioids
2- Abstract should contain the details for the optimal storage conditions as a final conclusion for this study.
3- In the abstract, novelty over related article should be stated. Besides, agreement or disagreement in short term stability results should be illustrated.
Ref 10 in this study
It seems that the conclusion for this study is different from the recently published article [10].
[Our experimental data proved that there is no need to extract the DBS devices and analyze the extracts immediately, since the targeted nitazenes show great stability once the collected blood droplet has been dried. Thus, storage of the DBS cards for several weeks at room temperature do not apparently pose the risk of analytes degradation. ]
4- In the introduction, line 69 and 70, correct the stamen [nitazenes pharmacokinetics is poorly investigated, increasing the difficulty of their 69 identification in real intoxication cases due to the lack of knowledge of the most appropri-70 ate biomarkers. ] as it is well investigated in male rats
Walton, S.E., Krotulski, A.J., Glatfelter, G.C., Walther, D., Logan, B.K. and Baumann, M.H., 2023. Plasma pharmacokinetics and pharmacodynamic effects of the 2-benzylbenzimidazole synthetic opioid, isotonitazene, in male rats. Psychopharmacology, 240(1), pp.185-198.
5- The aim of this study could be extend to illustrate the optimal storage conditions for DBS of nitazenes analogues
6- In 2.1. Method optimization and validation; it seems that different stationary phase and mobile system were used. So, the novelty over old LC-HRMS/MS methods should be highlighted more in section 2.1 and the abstract. This will enrich novelty and value for this article.
7- Line 117, write full name for HESI; line 119, NCE
8- In Table 1 , The values in table 1 for LOD and LOQ should be revised. In section 4.6.4; a common approach is to set a higher S/N threshold for LOQ, such as S/N ≥ 10. This ensures a more reliable quantification. As the authors mention in line 320, with a signal-to-noise ratio ≥ 3 was defined as LOD.
9- In figure 5 and 6, add number of sample in each analyzed concentration. Besides, there are unexpected increase of many concentrations such as etazene 5 ng/ml and flunitazene 5 ng/ml in fig 5 and isotonitazene 1 ng/ml. this is may related to personnel or instrumental errors . explanations are required.
10- In line 305, correct [Method linearity was evaluated by assessing 5 calibration curves in the range 1-20 ng/mL on 5 different days,] usually linearity was assessed in the same day.
11- Add reference for The Dixon test (p < 0.05)
12- Future plans should be extended more in the conclusion
Author Response
Comment 1: The authors assessed the stability of four opioids nitazenes (etazene, flunitazene, isotonitazene and protonitazene) in dried blood spot samples at different storage temperatures (room temperature and 4°C), using a novel well-validated liquid chromatography-high resolution mass spectrometry method. The work is well appreciated. The article is well written in a clear language. However, many revisions are required. Particularly, the authors should highlights novelty in LC-HR-MS/MS in the abstract. Besides, different conclusions should be heighted more if compared to outcomes of ref [10].
Response 1: we would like to thank the reviewer for the constructive comments, according that the manuscript was revised.
comment 2: Title should contain the word psychoactive opioids
Response 2: Amended
comment 3: Abstract should contain the details for the optimal storage conditions as a final conclusion for this study.
Response 3: As requested, a statement on the optimal storage conditions has been added in conclusion section (lines 33-37)
comment 4: In the abstract, novelty over related article should be stated. Besides, agreement or disagreement in short term stability results should be illustrated. Ref 10 in this study. It seems that the conclusion for this study is different from the recently published article [10]. [Our experimental data proved that there is no need to extract the DBS devices and analyze the extracts immediately, since the targeted nitazenes show great stability once the collected blood droplet has been dried. Thus, storage of the DBS cards for several weeks at room temperature do not apparently pose the risk of analytes degradation. ]
Response 4: We would like to thank the reviewer for the comment. Since the Journal guidelines report that the abstract should be a total of about 200 words maximum, we concisely summarized our results and their comparison to the suggested study. Unfortunately, the comparative analysis resulted difficult due to the different approach for stability assessment and the partial reported data (lines 19-24 and 33-37)
Comment 5: In the introduction, line 69 and 70, correct the stamen [nitazenes pharmacokinetics is poorly investigated, increasing the difficulty of their 69 identification in real intoxication cases due to the lack of knowledge of the most appropri-70 ate biomarkers. ] as it is well investigated in male rats (Walton, S.E., Krotulski, A.J., Glatfelter, G.C., Walther, D., Logan, B.K. and Baumann, M.H., 2023. Plasma pharmacokinetics and pharmacodynamic effects of the 2-benzylbenzimidazole synthetic opioid, isotonitazene, in male rats. Psychopharmacology, 240(1), pp.185-198.)
Response 5: In agreement with the reviewer, the sentence has been revised at lines 74-78
Comment 6: The aim of this study could be extend to illustrate the optimal storage conditions for DBS of nitazenes analogues
Response 6: Amended (lines 106-110)
Comment 7: In 2.1. Method optimization and validation; it seems that different stationary phase and mobile system were used. So, the novelty over old LC-HRMS/MS methods should be highlighted more in section 2.1 and the abstract. This will enrich novelty and value for this article.
Response 7: In agreement with the reviewer's suggestion, we highlighted in section 2.1 the novelty of the new method. Unfortunately, due to the limited number of words (n=200 max), the assessed conditions were just mentioned in the abstract. (Lines 116-133)
Comment 8: Line 117, write full name for HESI; line 119, NCE
Response 8: Amended
Comment 9: In Table 1 , The values in table 1 for LOD and LOQ should be revised. In section 4.6.4; a common approach is to set a higher S/N threshold for LOQ, such as S/N ≥ 10. This ensures a more reliable quantification. As the authors mention in line 320, with a signal-to-noise ratio ≥ 3 was defined as LOD.
Response 9: We would like to thank the reviewer for the comment and we apologize for the typo reported in the table 1. To this concern, the Table 1 values were controlled again and amended.
Comment 10: In figure 5 and 6, add number of sample in each analyzed concentration. Besides, there are unexpected increase of many concentrations such as etazene 5 ng/ml and flunitazene 5 ng/ml in fig 5 and isotonitazene 1 ng/ml. this is may related to personnel or instrumental errors . explanations are required.
Response 10: In agreement with the reviewer's comment, the number of samples has been added to the figures’ titles. Regarding the apparent concentration increase observed in the figures, our results showed a high variability of ratios in some sample groups. Personnel errors, instrumental errors, or interference from the ISTD can be excluded because the method had satisfactory validation parameters according to OSAC guidelines criteria. A possible reason for the varied concentrations of some points may be the hematocrit effect. Further studies on the hematocrit effect on nitazenes stability with different DBS devices need to be performed. These considerations were pointed out in the Discussion section (lines 243-252).
Comment 11: In line 305, correct [Method linearity was evaluated by assessing 5 calibration curves in the range 1-20 ng/mL on 5 different days,] usually linearity was assessed in the same day.
Response 11: Amended (lines 332-333)
Comment 12: Add reference for The Dixon test (p < 0.05)
Response 12: Amended
Comment 13: Future plans should be extended more in the conclusion
Response 13: Future plans were reported in the conclusion section (lines 381-388)
Round 2
Reviewer 2 Report
Comments and Suggestions for Authors
I do not see a considerable improvement of the manuscript. In addition to the objections mentioned in the first referee report, I think that two points increased the variability of the method: First of all, a punch was not used to get a reproducible sample, and furthermore it was show that a presoaking of the samples with water is required to obtain reproducible extraction results: https://doi.org/10.2144/btn-2019-0043
Reviewer 3 Report
Comments and Suggestions for Authors
The authors did all required recommendations. I appreciate their responses. The paper could be published in the current form.